# Scalable Neural Learning for Verifiable Consistency with Temporal Specifications

## Abstract

Formal verification of machine learning models has attracted attention recently, and significant progress has been made on proving simple properties like robustness to misclassification under perturbations of the input features. In this context, it has also been observed that folding the verification procedure into training makes it easier to train verifiably robust models. In this paper, we extend the applicability of verified training by extending it to (1) recurrent neural network architectures and (2) complex specifications that go beyond simple adversarial robustness, particularly specifications that capture temporal properties like requiring that a robot periodically visits a charging station or that a language model always produces sentences of bounded length. Experiments show that while models trained using standard training often violate desired specifications, our verified training method produces models that both perform well (in terms of test error or reward) and can be shown to be provably consistent with specifications.

## 1 Introduction

While deep neural networks (DNNs) have shown immense progress on diverse tasks (Sutskever et al., 2014; Mnih et al., 2015; Silver et al., 2016), they are often deployed without formal guarantees of their correctness and functionality. Their performance is typically evaluated using test data, or sometimes with adversarial evaluation (Carlini & Wagner, 2017; Uesato et al., 2018; Ebrahimi et al., 2018; Wang et al., 2019). However, such evaluation does not provide formal guarantees regarding the absence of rare but possibly catastrophic failures (Administration; Board; Ross & Swetlitz, 2018).

Researchers have therefore started investigating formal verification techniques for DNNs. Most of the focus in this direction has been restricted to feedforward networks and robustness to adversarial perturbations (Tjeng et al., 2017; Raghunathan et al., 2018b; Ko et al., 2019). However, many practically relevant systems involve DNNs that lead to sequential outputs (e.g., an RNN that generates captions for images, or the states of an RL agent). These sequential outputs can be interpreted as real-valued, discrete-time signals. For such signals, it is of interest to provide guarantees with respect to temporal specifications (e.g., absence of repetitions in a generated sequence, or that a generated sequence halts appropriately). Temporal logic provides a compact and intuitive formalism for capturing such properties that deal with temporal abstractions.

Here, we focus on Signal Temporal Logic (STL) (Donzé & Maler, 2010) as the specification language and exploit its quantitative semantics to integrate a verification procedure into training to provide guarantees with regard to temporal specifications. Our approach builds on recent work (Mirman et al., 2018; Gowal et al., 2018), which is based on propagating differentiable numerical bounds through DNNs, to include specifications that go beyond adversarial robustness. Additionally, we propose extensions to Mirman et al. (2018); Gowal et al. (2018) that allow us to train auto-regressive GRUs/RNNs to certifiably satisfy temporal specifications. We focus on the problem of verified training for consistency rather than post-facto verification. To summarize, our contributions are as:

- We present extensions to Mirman et al. (2018); Gowal et al. (2018) that allow us to extend verified training to novel architectures and specifications, including complex temporal specifications. To handle the auto-regressive decoder often used in RNN-based systems, we leverage differentiable approximations of the non-differentiable operations.

- We empirically demonstrate the applicability of our approach to ensure verifiable consistency with temporal specifications while maintaining the ability of neural networks to achieve high accuracy on the underlying tasks across domains. For supervised learning, verified training on the train-data enables us to provide similar verification guarantees for unseen test-data.

- We show that verified training results in robust DNNs whose specification conformance is significantly easier to guarantee than those trained adversarially or with data augmentation.

## 2 RELATED WORK

Here, we discuss the most closely related approaches covering AI safety and DNN verification.

### 2.1 NEURAL NETWORK VERIFICATION

There has been considerable recent progress on developing techniques for neural network verification, starting with the pioneering work of Katz et al. (2017), where a satisfiability-modulo-theories (SMT) solver was developed to verify simple properties for piecewise linear deep neural networks. Subsequently, several mature solvers that rely on combinatorial search (Tjeng et al., 2017; Dutta et al., 2017; Bunel et al., 2018) have helped scale these techniques to larger networks.

More recently, verification of neural networks using incomplete over-approximations – using dual based approaches (Dvijotham et al., 2018), and propagating bounds through the neural network (Gehr et al., 2018; Wang et al., 2018c; Weng et al., 2018a; Singh et al., 2019) – has emerged as a more scalable alternative for neural network verification. Raghunathan et al. (2018a); Wong et al. (2018) showed that folding the verification procedure into the training loop (called verified training) enables us to obtain stronger guarantees. Building on this line of work, Gowal et al. (2018) showed that training with simple interval bounds, with carefully chosen heuristics, is an effective approach towards training for verifiability. However, the focus of the above mentioned works on verified training is limited to adversarial robustness properties, and feedforward networks with monotonic activation functions. In this work, we build on Mirman et al. (2018); Gowal et al. (2018) to consider richer specifications that capture desired temporal behavior, and novel architectures with non-differentiable components. Table 1 compares different methods developed for verified training.

Table 1: Comparison of methods developed for training for consistency with specifications.

|  | Beyond ReLU | Continuous Input Features | Components with Gating | Auto-regressive | Temporal Specifications |
|---|---|---|---|---|---|
| Raghunathan et al. (2018a); Wong et al. (2018) Wang et al. (2018b) |  | ✓ |  |  |  |
| Gowal et al. (2018); Mirman et al. (2018) | ✓ | ✓ |  |  |  |
| Ghosh et al. (2018); Xiao et al. (2018) Li et al. (2017a) | ✓ |  | ✓ | ✓ | ✓ |
| Ours | ✓ | ✓ | ✓ | ✓ | ✓ |

In independent and concurrent work, Jia et al. (2019) develop an approach for verifying LSTMs, CNNs and networks with attention-mechanism. They use a similar approach as developed in our paper to compute bounds through the softmax function, word-substitutions, and also extend bound-propagation to handle the gating mechanism. Their main focus is robustness to misclassification. In contrast, we consider complex temporal specifications, and auto-regressive architectures.

### 2.2 SATISFACTION OF TEMPORAL PROPERTIES

**Temporal Specifications** While training networks to satisfy temporal logic specifications has been considered before, it has largely been from the perspective of encouraging RL agents to do so through a modified reward (Icarte et al., 2018b; Aksaray et al., 2016; Hasanbeig et al., 2018; Icarte et al., 2018a; Sadigh et al., 2014; Wen et al., 2017; Li et al., 2017a). The temporal logic specification is used to express the task to be performed by the agent, rather than as a verifiable property of the system. Ghosh et al. (2018) encourage specification conformance during training by regularizing with a loss arising from the desired specification. However, the specification is enforced on specific inputs and does not guarantee that the property holds across continuous regions of the input space (e.g., all

inputs in the neighborhood of a given image). In contrast, we train DNNs to verifiably satisfy rich temporal specifications over large/continuous sets of inputs. Further, we note that there is work on falsifying STL specifications using stochastic optimization(Annpureddy et al., 2011; Donzé, 2010), however our focus here is on verified training.

**Safe RL** Safe exploration methods (Garcıa & Fernández, 2015) consider explorations that do not visit unsafe states. Temporal logics, in general, permit richer specifications of desired temporal behaviors than avoiding unsafe states. Junges et al. (2016) synthesize a scheduler that limits the agent explorations to safe regions of the environment specified using probabilistic computation tree logic (PCTL). An alternative mechanism, called shields, monitors actions of an agent and restricts it to a safe subset ensuring conformance to specifications in linear temporal logic (Alshiekh et al., 2018) or PCTL (Jansen et al., 2018). Instead of using external mechanisms to restrict the agent choices, we incorporate temporal logic objectives into training and achieve verified training of our agents. Furthermore, our work is not restricted to training verifiable RL agents. We demonstrate the generality of our approach on image captioning and language generation tasks involving RNNs.

**Verification using Interpretable Policies** PIRL (Verma et al., 2018) and VIPER (Bastani et al., 2018) extract interpretable policies that are also amenable to verification. These approaches first learn DNN agents and then use imitation learning to extract interpretable/verifiable policies. Wang et al. (2018a) analyze RNNs by distillation to a Deterministic Finite Automaton (DFA), and demonstrate successful distillation on recognizing Tommika Grammar. However, this distillation is often difficult, and remains an open challenge for vision/language processing tasks. Further, there are no guarantees about the original DNN. In contrast, we focus on guarantees for the DNN itself.

# 3 Formulating Temporal Constraints on ML Models with Signal Temporal Logic

We consider the problem of verified training of a model with respect to a desired property. In what follows, we describe how signal temporal logic provides a formalism to describe properties of interest.

## 3.1 Preliminaries

We use Signal Temporal Logic (STL) (Donzé & Maler, 2010), an extension of Linear Temporal Logic (LTL) (Pnueli, 1977) that can reason about real-valued signals.

**Syntax and Qualitative Semantics** STL has the following syntax:

$$\varphi := \mathtt{true} \,|\, q(s) \geq 0 \,|\, \neg\varphi \,|\, \varphi_1 \wedge \varphi_2 \,|\, \varphi_1 \mathcal{U}_I \varphi_2 \tag{1}$$

where $\mathtt{true}$ is the Boolean constant for truth, and $\neg$ and $\wedge$ are Boolean negation and conjunction operators. The symbol $q$ is a quantifier-free non-linear real-valued arithmetic function over the vector-valued state denoted by $s$; the formula $q(s) \geq 0$ is called an atom. The formula $\varphi_1 \mathcal{U}_I \varphi_2$ is the *until* temporal operator, meaning $\varphi_1$ holds until $\varphi_2$ holds in the interval $I$. We define $\Diamond_I \varphi$ (meaning $\varphi$ *eventually* holds in the interval $I$) as $\mathtt{true}\,\mathcal{U}_I \varphi$ and $\Box_I \varphi$ (meaning $\varphi$ *always* holds in $I$) as $\neg\Diamond_I \neg\varphi$.

In this work, we interpret STL formulae over a trace $\sigma$ which is a discrete-time, vector-valued signal; $\sigma_t$ denotes the value of the signal at time $t$. We write $(\sigma, t) \models \varphi$ to indicate that $\varphi$ holds for $\sigma$ at time $t$. An atom $q(s) \geq 0$ holds at time $t$ if $q(\sigma_t) \geq 0$. Trivially, $(\sigma, t) \models \mathtt{true}$ always holds. An until formula $\varphi_1 \mathcal{U}_I \varphi_2$, with $I = [a, b]$, holds at a time instance $t$ if $(\sigma, t') \models \varphi_2$ for some time $t' \in [t + a, t + b]$ and $(\sigma, t'') \models \varphi_1$ for all $t'' \in [t, t']$. In the rest of the paper, we use $t + I$ to denote $[t + a, t + b]$ for $I = [a, b]$. A formula $\varphi$ is said to be *satisfied* over a trace $\sigma$ if $(\sigma, 0) \models \varphi$. In this work, we restrict $I$ to be bounded-intervals of time. We refer the reader to Donzé & Maler (2010) for a detailed introduction to the semantics of STL specifications.

While bounded-time STL properties can be unrolled through time into logical properties using the Boolean conjunction and disjunction operators (Raman et al., 2015), STL provides a succinct and intuitive notation for expressing desired temporal properties. In contrast with prior work on verified training that only considers adversarial robustness (a linear constraint on the logits), we consider general specifications that assert temporal properties over input-output behaviors of neural networks. Section 3.2 lists several examples of relevant properties that can be expressed in bounded-time STL.

## 3.2 STL Specifications for Learning Tasks

To illustrate our approach, we consider three temporal properties that we want our networks to satisfy.

### 3.2.1 Bounding Caption Length for Image-Captioning

Multi-MNIST images consist of non-overlapping MNIST digits on a canvas of fixed size (Figure 1). The number of digits in each image varies between 1 and 3. The task is to label the sequence of digits in the image, followed by an end of sequence token. Prior work on this task (Wang et al., 2019) has shown image-to-sequence models to be vulnerable to generating sequences longer than the true number of digits in the image, under small adversarially chosen perturbations. Here, we consider the task of training a DNN that does not output sequences longer than the desired length, while achieving similar nominal task performance.

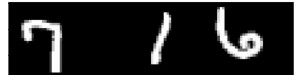

Figure 1: MMNIST Image

Let $y := f(x)$ be the sequence of logits output by the RNN model when given input image $x$. For an image $x$, the termination specification is formalized as follows:

$$\forall \Delta x \in \{s : \|s\|_\infty \leq \epsilon\}.(f(x + \Delta x), 0) \models \varphi_x, \tag{2}$$

where $\varphi_x(y) := \Diamond_{[0,t_x^*]} \bigwedge_{i \neq e} (y[t]_e - y[t]_i) \geq 0$, $t_x^*$ is the true number of digits in the image $x$, $e$ is the label corresponding to the end of sequence token, $\epsilon > 0$ is the perturbation bound. Informally, this specification enforces that the end of sequence token is output no later than after the true number of digits have been output by the RNN, for all inputs within $\epsilon$ distance from a true-image.

### 3.2.2 Verifying that a Robot Never Runs Out of Charge

To demonstrate our approach in the RL setting, we consider a task with a vacuum cleaning robot. We summarize this task here (See Appendix F for more details). The agent (robot) operates in a continuous domain with its location in $(x, y) \in [0, 25]^2$ (Figure 2, Appendix). The room is divided into discrete cells, and the agent gets a reward for visiting any "dirty" cell which has not been visited in the previous $T_{\text{dirt}}$ time-steps. The agent must visit one of the recharge cells every $T_{\text{recharge}}$ time-steps, or the episode is terminated with no further reward. The policy maps observations (of the agent location and a map of the room) to continuous velocity controls. We use $f_\theta$ to denote the result of applying the policy, parameterized by $\theta$, followed by the environment update.

For this agent, we want to verify the specification: $\forall z \in S_\epsilon.(f_\theta(z), 0) \models \Box_{[0,T]} \Diamond_{[0,T_{\text{recharge}}]} \varphi_{\text{recharge}}$, where $\varphi_{\text{recharge}}$ corresponds to the agent being in one of the recharge cells. This specification ensures that, for a set of feasible starting positions $S_\epsilon$, for every time-step $t$ in $[0, T]$, the agent recharges itself at least once within $T_{\text{recharge}}$ time-steps. See Appendix F.1 for a detailed description of $S_\epsilon$.

### 3.2.3 Verifying Generated Outputs from a Language Model

A common failure mode for language models is their tendency to fall into degenerate loops, often repeating a stop-word (Wang et al., 2019). To illustrate the applicability of STL specifications in this setting, we show how to formalize the property that a GRU language model does not repeat words consecutively. We call this specification bigram non-repetition. More concretely, the desired specification is that the output sequence does not contain bigram repetition amongst the 100 most frequent tokens in the training corpus vocabulary. We want to verify this property over a large set of possible conditioning inputs for the generative model. Concretely, we define an input set $S$ of roughly 25 million prefixes generated from a syntactic template (See Appendix G.1 for details). These prefixes are input to the LM, and then we evaluate the specification on the model output.

Now, consider a prefix $x$ and the sequence of logits $y$ output by the recurrent GRU network $f$ (i.e. $y = f(x)$), with $y(t)_k$ referring to the logit corresponding to the $k^{th}$ most-frequent token in the vocabulary at time $t$. A compact formal specification $\varphi_{\text{bigram}}$ ruling out bigram repetition is:

$$\varphi_{\text{bigram}} := \Box_{[0,T_{\text{sample}}]} \bigwedge_{i=1,2,\ldots,100} \left( \left( \bigwedge_{j \neq i} y(t)_i \geq y(t)_j \right) \rightarrow \Diamond_{[0,1]} \neg \left( \bigwedge_{j \neq i} y(t)_i \geq y(t)_j \right) \right) \tag{3}$$

where $T_{\text{sample}}$ denotes the length of the generated sample, in our case 10. The RNN $f$ is required to satisfy the specification $\forall x \in S.(f(x), 0) \models \varphi_{\text{bigram}}$.

## 4 VERIFIABLE DNN TRAINING FOR STL SPECIFICATIONS

We consider the problem of learning a trace-valued function $f_\theta$ to verifiably satisfy a specification of the form $\forall x \in S. (f_\theta(x), 0) \models \varphi$, where input $x$ ranges over set $S$, and $f_\theta(x)$ is the trace generated by $f_\theta$ when evaluated on $x$, $\theta$ represents the trainable parameters, and $\varphi$ is an STL specification. We drop $\theta$ for brevity, and simply denote $f_\theta(x)$ as $f(x)$. Formally, our problem statement is:

 *Given a set of inputs S, train the parameters $\theta$ of $f_\theta$ so that $\forall x \in S. (f_\theta(x), 0) \models \varphi$, where $\varphi$ is a bounded-time STL specification.*

### 4.1 OPTIMIZATION FORMULATION OF STL VERIFICATION

For an STL specification $\varphi$, its quantitative semantics can be used to construct a function $\rho(\varphi, f(x), t)$ whose scalar valued output is such that $\rho(\varphi, f(x), t) \geq 0 \iff (f(x), t) \models \varphi$ (Donzé & Maler, 2010). In terms of the quantitative semantics, the verification problem is equivalent to showing that $\forall x \in S. \rho(\varphi, f(x), 0) \geq 0$. This verification task can be written as the optimization problem of finding the sequence of inputs $x$ such that the sequence of outputs $f(x)$ result in the strongest violation of the specification with regard to the quantitative semantics:

$$\min_x \rho(\varphi, f(x), 0) \text{ subject to } x \in S. \tag{4}$$

If the solution to equation 4 is negative, then there exists an input leading to the violation of $\varphi$.

### 4.2 BOUND PROPAGATION

The optimization problem in equation 4 itself is often intractable; even in the case when the specification is limited to robustness against perturbations in a classification task, it is NP-hard (Katz et al., 2017). There are tractable approaches to bounding the problem in equation 4 (Raghunathan et al., 2018a; Dvijotham et al., 2018), but the bounds are often too loose to provide meaningful guarantees. To obtain a tighter bound tractably, interval bound propagation – which by itself provides loose bounds, but is efficient to compute (2x computational cost) – can be leveraged for verified training to give meaningful bounds on robustness under $l_\infty$ perturbations (Mirman et al., 2018; Gowal et al., 2018). Our general approach for doing bound propagation on the function $f$ is to use standard interval arithmetic. While this is straightforward when $f$ is a feedforward DNN (Gowal et al., 2018), here we extend bound propagation to a richer set of (temporal) specifications and architectures. First, we highlight the novel aspects of bound propagation required for (a) auto-regressive RNNs/GRUs, (b) STL specifications.

**Bound Propagation through GRUs**   Computing bounds across GRU cells involves propagating bounds through a multiplication operation (as a part of gating mechanisms), which can be handled by a straightforward application of interval arithmetic (Hickey et al., 2001) (see Appendix H).

**Bound Propagation through auto-regressive RNNs**   For language modeling and image captioning, we use GRU decoders with greedy decoding. Greedy-decoding involves a composition of the `one-hot` and the `argmax` operations. Both of these operations are non-differentiable. To overcome this and compute differentiable bounds (during training), we approximate this composition with a `softmax` operator (with a low temperature $T$). In the limit, as $T \to 0$, the softmax operator converges to the composition `one-hot(argmax(·))` For propagating bounds through the softmax operator, we leverage that the bounds are monotonic in each of the individual inputs. Formally, given a lower ($\underline{p}$) and upper ($\overline{p}$) bound on the input $p$ to a softmax layer (i.e., $\underline{p} \leq p \leq \overline{p}$), the lower $\underline{w}$ and upper bound($\overline{w}$) on the output can be computed as:

$$\overline{s} = \sum_{i=1}^{N} \exp \overline{p}_i, \quad \underline{s} = \sum_{i=1}^{N} \exp \underline{p}_i, \quad \Delta_i = \exp \overline{p}_i - \exp \underline{p}_i, \quad \overline{w}_i = \frac{\exp \overline{p}_i}{\underline{s} + \Delta_i} \quad \underline{w}_i = \frac{\exp \underline{p}_i}{\overline{s} - \Delta_i},$$

where $p_i$ is the $i^{th}$ coordinate of $p$ and $p \in \mathbb{R}^N$. During evaluation, the `one-hot(argmax(.))` function is used as is. Given bounds on each coordinate of $p$ (i.e., $\underline{p} \leq p \leq \overline{p}$) and $s =$

one-hot(argmax($p$)), bounds on coordinate $s_i$ can be computed as:

$$\underline{s}_i(x) = \begin{cases} 1 & \underline{p}_i \geq \overline{p}_j. \forall j \\ 0 & \text{otherwise.} \end{cases} \qquad \overline{s}_i(x) = \begin{cases} 0 & \exists j \neq i \text{ such that, } \underline{p}_j > \overline{p}_i \\ 1 & \text{otherwise.} \end{cases}$$

We discuss bound propagation for discrete inputs in Appendix C

**Bound Propagation through the specification** First, we extend the quantiative semantics for STL specifications (Donzé & Maler, 2010) to allow us to reason over sets of inputs. For a STL specification $\varphi$ in negation normal form (NNF) (See Appendix A for details on the quantitative semantics and conversion to NNF), we first define a *lower bound* for the quantitative semantics of $\varphi$ over the set $S$, which we denote by $\omega_{S,f}(\varphi, 0)$. We define this bound assuming we have lower bounds on all the atoms occurring in $\varphi$. Specifically, let $\Omega_{S,f}(q, t)$ be a lower bound on $q(f(x)_t)$ over all inputs $x \in S$; in other words, at each time $t$ we have $\forall x \in S. \Omega_{S,f}(q, t) \leq q(f(x)_t)$. Now, we define the lower bound on a specification $\varphi$ inductively as:

- $\omega_{S,f}(\texttt{true}, t) = +\infty, \qquad \omega_{S,f}(\neg\texttt{true}, t) = -\infty, \qquad \omega_{S,f}(q(s) \geq 0, t) = \Omega_{S,f}(q, t)$
- $\omega_{S,f}(\varphi_1 \wedge \varphi_2, t) = \min(\omega_{S,f}(\varphi_1, t), \omega_{S,f}(\varphi_2, t))$
- $\omega_{S,f}(\varphi_1 \vee \varphi_2, t) = \max(\omega_{S,f}(\varphi_1, t), \omega_{S,f}(\varphi_2, t))$
- $\omega_{S,f}(\varphi_1 \mathcal{U} \varphi_2, t) = \max\limits_{t' \in t+I} \min\left(\omega_{S,f}(\varphi_2, t'), \min\limits_{t'' \in [t,t']} \omega_{S,f}(\varphi_1, t'')\right).$

**Lemma 1.** *For any time $t$, given lower bounds $\Omega_{S,f}(q, t)$ on all the atoms $q(s) \geq 0$ in $\varphi$, we have:*
$$\forall x \in S. \omega_{S,f}(\varphi, t) \leq \rho(\varphi, f(x), t)$$

**Corollary 1.** *If $\omega_{S,f}(\varphi, t) \geq 0$, then $\forall x \in S. (f(x), 0) \models \varphi$.*

See Appendix B for proof of Lemma 1. In order to compute the lower bounds $\Omega_{S,f}(q, t)$ required for Lemma 1, given bounds on the input $x$, we can first compute bounds on the outputs $f(x)_t$ at each time $t$. For the atoms $q(s) \geq 0$ appearing in $\varphi$, given bounds on the input $s$ we can compute bounds on $q(s)$. These bounds can then be propagated through the specification inductively.

### 4.3 VERIFIED TRAINING FOR STL SPECIFICATIONS

In this section, we describe how to train a network to satisfy an STL specification $\varphi$. The quantitative semantics $\rho(\varphi, \sigma, 0)$ gives a degree to which $\sigma$ satisfies $\varphi$. First, we compute lower bounds on the values of the atoms in $\varphi$ at each instance of time. Then, by application of Lemma 1, we can compute the lower bound $\omega_{S,f}(\varphi, 0)$ satisfying $\forall x \in S. \omega_{S,f}(\varphi, 0) \leq \rho(\varphi, f(x), 0)$. Subsequently we optimize the lower bound $\omega_{S,f}(\varphi, 0)$ to be non-negative, thereby guaranteeing that the specification of interest holds: $\forall x \in S. \rho(\varphi, f(x), 0) \geq 0$.

Let $L_{obj}$ be the loss objective corresponding to the base task, for example, the cross-entropy loss for classification tasks. Training thus requires balancing two objectives: minimizing loss on the base task by optimizing $L_{obj}(f_\theta)$, and ensuring the positivity of $\omega_{S,f_\theta}$. We can use gradient descent to directly optimize the joint loss: $L_{obj}(f_\theta) - \lambda \min\{\omega_{S,f_\theta}(\varphi, 0), \tau\}$, where $\lambda$ is a scalar hyper-parameter, $\tau$ is a positive scalar threshold ($\tau \in \mathbb{R}_+$). The clipping avoids having to carefully balance the two losses. The quantitative semantics of an STL specification $\varphi$ is a non-smooth function of the weights of the neural network, and is difficult to optimize directly with gradient descent. We find in practice that curriculum training, similar to Gowal et al. (2018), works best for optimizing the specification loss, starting with enforcing the specification over a subset $S' \subset S$, and gradually covering the entire $S$. Empirically, the curriculum approach means that the task performance ($L_{obj}$) does not degrade much.

## 5 EXPERIMENTAL RESULTS

### 5.1 SEQUENTIAL CAPTIONING OF MULTI-MNIST IMAGES

For this task, we perform verified training to enforce the termination specification $\varphi_x$ (equation 4) on the training data as discussed in Section 4.3. Post training, for unseen test-set images, we evaluate the quantitative specification loss $\omega_{S_{x,\epsilon},f}(\varphi_x, 0)$. For an image $x$ from the test-set, if $\omega_{S_{x,\epsilon},f}(\varphi_x, 0)$ is positive, it is guaranteed that there is no input within an $l_\infty$ radius of $\epsilon$ around the current image that can cause the RNN to generate a longer sequence than the number of true digits in the image.

Table 2: Comparison of GRU training methods on the MMNIST task. We evaluate against the termination specification on different metrics, and also report nominal accuracy. '–' indicates a trivial verified accuracy of $0\%$ obtained with bound propagation. The entries with *Verified Termination Accuracies* corresponding to $0.0$ are those where we were able to generate adversarial examples (counter-examples) to the specification for every point in the test-set. We found that adversarial training is difficult because of the presence of the sigmoid & tanh activation functions commonly used in GRUs. To have a meaningful baseline, we performed adversarial training on an RNN (feedforward cells with ReLU activation). For $\epsilon = 0.1$, attacking the loss from Wang et al. (2019) to produce longer sequences performs better, while for the other $\epsilon$ values adversarial training with the STL quantitative loss performs better. Adversarial training performs well but is difficult to verify. At larger $\epsilon$, verified training results in both better guarantees (specification conformance), and better nominal accuracies.

| Perturbation $\epsilon$ | Training | Nominal Accuracy | Verified Termination Accuracy | Adversarial Termination Accuracy |
|---|---|---|---|---|
| | Verifiable | 94.9 | **98.3** | **100.0** |
| 0.1 | Adversarial | 94.1 | – | **100.0** |
| | Nominal | **95.9** | – | 33.5 |
| | Verifiable | 94.5 | **98.7** | **100.0** |
| 0.2 | Adversarial | 93.3 | – | **100.0** |
| | Nominal | **95.9** | – | 20.94 |
| | Verifiable | 94.4 | **98.7** | **100.0** |
| 0.3 | Adversarial | 90.0 | – | 99.7 |
| | Nominal | **95.9** | 0.0 | 0.0 |
| | Verifiable | 94.1 | **99.0** | **100.0** |
| 0.5 | Adversarial | 75.6 | – | 100.0 |
| | Nominal | **95.9** | 0.0 | 0.0 |

In Tables 2 and 3, *verified termination accuracy* refers to the fraction of unseen data for which we can verify the absence of counter-examples to the termination property (equation 4). *Nominal accuracy* refers to the percentage of correctly predicted tokens – including the end of sequence token. Table 2 compares verified training with nominal and adversarial training. Verified training outperforms

Table 3: We train the RNN with ReLU activations from (Wang et al., 2019) to be verifiable with $\epsilon = 0.3$, and compare its verifiability with MILP based verification reported in Wang et al. (2019) at different perturbation radii. The nominal accuracy for the model trained to be verifiable is $93.9\%$ and model trained in a standard manner is $96.4\%$. For larger perturbations, the MILP solver times out. '–' indicates that we were unable to certify robustness for any of the points in the test-set, for the given perturbation within the time-out window of 30 minutes.

| Perturbation Radius $\epsilon$ | Training | Verification Method | Verified Termination Accuracy |
|---|---|---|---|
| 0.002 | Nominal | MILP | 83.00 |
| | Verifiable | Bound Prop. | **99.01** |
| 0.02 | Nominal | MILP | – |
| | Verifiable | Bound Prop. | **98.95** |
| 0.3 | Nominal | MILP | – |
| | Verifiable | Bound Prop. | **94.3** |

both adversarial and nominal training on both adversarial and verified termination accuracy metrics. The pixel values are scaled to be in the range $[0, 1]$. At perturbations of size $\epsilon = 0.5$, the images can be turned gray; however, the DNN remains robust to such large perturbations by predicting that the image has no more than a single digit at large perturbations, while maintaining nominal accuracy on clean data. This in contrast with robustness against misclassification, where it is not possible to be robust at large perturbations because the specifications for images from different classes conflict. Adversarial accuracy is evaluated with the iterative attack from Wang et al. (2019) (10000 steps).

**Run-time Considerations**   As another baseline, we compare with verified termination accuracies from Wang et al. (2019)(Table 3). In Wang et al. (2019), the greedy-decoding and the specification are turned into a MILP-query solved with the SCIP solver (Gleixner et al., 2018). Further, we use ReLU RNNs here because GRUs are not amenable to MILP solvers. Verified training allows us to certify specification conformance for much larger perturbations ($\approx 2$ orders of magnitude larger).

## 5.2   An RL Mobile-Robot Agent

We consider the recharging specification $\varphi_{\text{recharge}}$ over a time-horizon of $T = 10$, for an agent starting within a $l_\infty$ distance of $\epsilon$ from the center of the any of the cells (See Appendix F.1 for

details on feasible initial states). To regularize the DNN to be verifiable with regard to $\varphi_{\text{recharge}}$, the specification loss is obtained by rolling out the current policy through time, and propagating bounds through the rolled out policy and the dynamics. This assumes a deterministic dynamics model.

We compare our verifiably trained agent to both a vanilla RL agent, and an agent trained with reward shaping as in Li et al. (2017b). All agents achieve a similar reward, and we do not find specification violations for roll-outs from $10^6$ random (feasible) initial states. To compare verifiability, we discretize a region within a distance of $\epsilon$ to each cell-center into $10^2$ $l_\infty$ balls, and verify with bound-propagation that the agent satisfies $\varphi_{\text{recharge}}$ for each sub-region. Agents trained with verified training are significantly more verifiable than agents trained otherwise, with little degradation in performance (Table 4), which is consistent with prior work in classification (Wong & Kolter, 2018)

Table 4: Mean/Variance performance (across 5 agents of each type) across different metrics. For each agent, reward is computed as mean across 100 episodes. $\epsilon$ is distance from the center of the grid cells, and for each $\epsilon$ we report the fraction of the cells for which we are able to certify that $\varphi_{\text{recharge}}$ holds.

| Training | % of cells verified ($\epsilon = 1.0$) | % of cells verified ($\epsilon = 0.1$) | % of cells verified ($\epsilon = 0.01$) | % of cells verified ($\epsilon = 0.001$) | % of cells verified ($\epsilon = 0.0001$) | Reward |
|---|---|---|---|---|---|---|
| Verifiable | 100.0/0.0 | 100.0/0.0 | 100.0/0.0 | 100.0/0.0 | 100.0/0.0 | 12.71/0.19 |
| Standard | 15.8/9.0 | 64.9/6.8 | 77.6/5.3 | 90.3/1.8 | 99.2/0.0 | 12.85/0.06 |
| Reward Shaping | 39.1/21.9 | 74.3/8.6 | 83.2/5.9 | 92.0/1.9 | 100.0/0.0 | 12.76/0.22 |

## 5.3 Language Generation

Our language model consists of a 2-layer GRU with 64 hidden nodes per layer, trained on the tiny Shakespeare corpus using a word embedding dimension of 32, and vocabulary truncated to the 2500 most frequent training words. We evaluate the model's ability to satisfy $\varphi_{\text{bigram}}$. We compare both a nominal model trained using log-likelihood, a model that randomly samples prefixes from the input space and penalizes violations to the specification, and verified training that covers the full input space (Details in Appendix G). We report test set perplexity and count of violations observed over the 25M prefixes (Table 5).

Table 5: Language model perplexity, number of failures during an exhaustive enumerative search over the 25M perturbations, and computational cost of verification (number of forward passes).

| Training | Perplexity | # Failures | # Verification Cost |
|---|---|---|---|
| Verifiable | 228.91 | **0** | $\approx$**2** |
| Sampled | 174.89 | **0** | $2.57 \times 10^7$ |
| Nominal | **153.63** | $1.79 \times 10^7$ | $2.57 \times 10^7$ |

We find that while standard training achieves the best perplexity results, it also produces numerous specification failures. Sampling prefixes and regularizing them to avoid bigram repetition using $\rho(\varphi_{\text{bigram}}, f(x), 0)$ eliminates failures, but the overall evaluation cost of the exhaustive search is large. Verifiable training with bound propagation, by contrast, comes with a constant computational cost of $\approx 2$ forward passes. This is because matrix multiplications form a significant majority of the computational cost during a forward pass, and propagating bounds through a layer of the form $y = \sigma(Wx + b)$, where $\sigma$ is a monotonic activation function (e.g. ReLU, sigmoid, tanh), can be performed such that it only costs twice as much as a normal forward pass (Gowal et al., 2018).

**Run-time Considerations** Verification with propagating bounds can be performed in under **0.4 seconds** (including propagating bounds through the spec), while exhaustive search over 25M prefixes for specification violations takes over **50 minutes**. Further, as possible word substitutions increase, the cost for exhaustive search grows exponentially while that for bound propagation stays constant.

## 6 Conclusion

Temporal properties are commonly desired from DNNs in settings where the outputs have a sequential nature. We extend verified training to tasks that require temporal properties to be satisfied, and to architectures such as auto-regressive RNNs whose outputs have a sequential nature. Our experiments suggest that verified training leads to DNNs that are more verifiable, and often with fewer failures.

Future work includes extending verification/verified training to unbounded temporal properties. Another important direction is to develop better bound propagation techniques that can be leveraged for verified training. In the RL setting, an important direction is data-driven verification in the absence of a known model of the environment.

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

# Appendices

## A   STL QUANTITATIVE SEMANTICS

In addition to the qualitative semantics discussed in the main text, STL formulae have quantitative semantics (Donzé et al., 2013; Donzé & Maler, 2010) defined inductively by the function $\rho$ below. For a given trace $\sigma$, with $\sigma_t$ indicating the value of the signal at time $t$, the quantitative semantics is given by:

- $\rho(\texttt{true}, \sigma, t) = +\infty$

- $\rho(q(s) \geq 0, \sigma, t) = q(\sigma_t)$

- $\rho(\neg\varphi, \sigma, t) = -\rho(\varphi, \sigma, t)$

- $\rho(\varphi_1 \wedge \varphi_2, \sigma, t) = \min\left(\rho(\varphi_1, \sigma, t), \rho(\varphi_2, \sigma, t)\right)$

- $\rho(\varphi_1 \, \mathcal{U}_I \, \varphi_2, \sigma, t) = \max\limits_{t' \in t+I} \min\left(\rho(\varphi_2, \sigma, t'), \min\limits_{t'' \in [t,t']} \rho(\varphi_1, \sigma, t'')\right)$

One can obtain the qualitative semantics from the sign of the quantitative semantics. Specifically, $(\sigma, t) \models \varphi \iff \rho(\varphi, \sigma, t) \geq 0$.

We can convert the formulae to their equivalent negation normal form by following the standard procedure until negations are only associated with atoms and Boolean constants. In particular, we interpret $\rho(\neg\texttt{true}, \sigma, t) = -\infty$ and use the disjunction operator defined as $\rho(\varphi_1 \vee \varphi_2, \sigma, t) = \max(\rho(\varphi_1, \sigma, t), \rho(\varphi_2, \sigma, t))$. The normal form for $\neg(\varphi_1 \mathcal{U}_I \varphi_2)$ is obtained by pushing the negation into the subformulae, and swapping $\min$ with $\max$. Finally, we turn $\neg(q(s) \geq 0)$ into $-q(s) > 0$ which we approximate by $-q(s) - \delta \geq 0$ for some small $\delta > 0$.

## B   PROOF OF LEMMA 1

*Proof.* We proceed by induction on $\varphi$. The base cases and the conjunction case are straightforward, and the atom case follows by assumption. The disjunction case requires us to show: $\omega_{S,f}(\varphi_1 \vee \varphi_2, t) \leq \min\limits_{x \in S}(\max(\rho(\varphi_1, f(x), t), \rho(\varphi_2, f(x), t)))$. Applying the max-min inequality, the right hand side is at least $\max(\min\limits_{x \in S}(\rho(\varphi_1, f(x), t)), \min\limits_{x \in S}(\rho(\varphi_2, f(x), t))$. Then using the inductive hypotheses, we know this is at least $\max(\omega_{S,f}(\varphi_1, t), \omega_{S,f}(\varphi_2, t))$, and the case follows. The case for the $\mathcal{U}$ operator has a similar proof based on the max-min inequality. $\square$

## C   BOUNDS FOR DISCRETE INPUTS

Tasks with discrete inputs, such as language generation tasks, encode a prefix sentence as conditioning before decoding a follow-up sequence of words. Consider prefixes of the form $x = x_0, x_1, \ldots$ such that $x_i \in S_i$, where $S_i$ is a finite set of tokens that can appear at position $i$ in the input sequence. We can propagate perturbations in the prefix by first projecting the tokens $S_i$ through the embedding layer $E$, and then considering the maximum and the minimum value along each embedding dimension to bound the output from $E$. Formally,

$$\underline{E}_j(x_i) = \min_{x_i \in S_i} E_j(x_i) \leq E_j(x_i) \leq \max_{x_i \in S_i} E_j(x_i) \leq \overline{E}_j(x_i). \tag{5}$$

Jia et al. (2019); Huang et al. (2019) also consider bound propagation for word substitutions.

| | Beyond ReLU | Training for Spec. Satisfaction | Continuous Input Features | Components with Gating | Auto-regressive components | Temporal Specifications |
|---|---|---|---|---|---|---|
| Dvijotham et al. (2018)
Zhang et al. (2018) | ✓ | ✗ | ✓ | ✗ | ✗ | ✗ |
| Raghunathan et al. (2018a); Wong et al. (2018)
Wang et al. (2018b) | ✗ | ✓ | ✓ | ✗ | ✗ | ✗ |
| Gowal et al. (2018); Mirman et al. (2018) | ✓ | ✓ | ✓ | ✗ | ✗ | ✗ |
| Dutta et al. (2017); Tjeng et al. (2017) | ✗ | ✗ | ✓ | ✗ | ✗ | ✗ |
| Ghosh et al. (2018); Xiao et al. (2018)
Li et al. (2017a) | ✓ | ✓ | ✗ | ✓ | ✓ | ✓ |
| Weng et al. (2018b) | ✗ | ✗ | ✓ | ✗ | ✗ | ✗ |
| Ko et al. (2019) | ✓ | ✗ | ✓ | ✓ | ✗ | ✗ |
| Ours | ✓ | ✓ | ✓ | ✓ | ✓ | ✓ |

Table 6: Comparison of methods developed for enforcing/checking consistency with specifications.

## D    DETAILED COMPARISON WITH RELATED WORK ON VERIFICATION/VERIFIED TRAINING

## E    MMNIST

Figure 1 depicts a sample image from the MMNIST dataset. For this task, for the GRU decoder, we train a 2 layer-convolution network with 32 filters in each layer, and the GRU decoder has 2 cells, each with a latent space of 64 dimensions. Further, to help training with the specification stabilize, we normalize the weights in the linear-layers of the GRU with the $l_1$ norm (similar to the Salimans & Kingma (2016), where the weights are normalized with the $l_2$-norm. For the curriculum training, we first train on the original task for 5000 steps to get the full clean accuracy, and then train with the specification loss for another 300000 steps to regularize the model to satisfy the specification. During this phase, we gradually ramp up the perturbation radius $\epsilon$ to the desired value, and also simultaneously increase the coefficient $\lambda$. The normalizing prevents the bounds from getting large during the wamring-up phase of training.

For adversarial training, we use a 7-step PGD attack from Madry et al. (2017), and we found that a similar curriculum helps stabilize adversarial training. Adversarial training with the largest perturbation radius as in Madry et al. (2017) degraded the performance on the nominal task signficiantly, while curriculum based adversarial training degrades performance to a much lesser extent

For the test-train data split on this task, we use the same as in Wang et al. (2019).

## F    RL AGENT

### F.1    INITIAL STATES OF THE RL AGENT

$S_\epsilon$ corresponds to the states $(x, y)$ within a $l_\infty$ distance of $\epsilon$ from the centre of each of the cells. We formally define this set below.

For a cell $i$ (for Figure 3, $i \in \{1, 2 \ldots 25\}$) with center $x_{c_i}, y_{c_i}$, the $\epsilon$-ball $S_{i,\epsilon}$ corresponds to the set of positions $(x_a, y_a)$ for the agent such that $\|(x_a - x_{c_i}, y_a - y_{c_i})\|_\infty \leq \epsilon$. Formally,

$$S_{i,\epsilon} := \{(x_a, y_a) : \|(x_a - x_{c_i}, y_a - y_{c_i})\|_\infty \leq \epsilon\}.$$

We can then define $S = \underset{i}{\cup} S_{i,\epsilon}$ as the set of feasible initial states of the agent for which we wish to verify the property. Table 4 reports the fraction of cells $i$ for which we are able to verify that the agent recharges on starting from $S_{i,\epsilon}$.

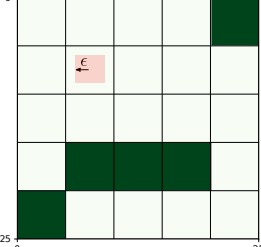

Figure 2: Domain for the robot. Recharge cells in green.

### F.2    TASK AND TRAINING DETAILS

The agent's observation at each time-step $t$ contains i) its own coordinates $(x_t, y_t)$, ii) for each cell, the time remaining until that cell is dirty, and iii)

the (fixed) locations of the recharge cells. The agent learns a policy from these observations to a continuous control action $(a_{x,t}, a_{y,t}) \in \mathbb{R}^2$. The continuous part of the agent's state is updated as:

$$x_{t+1} = x_t + a_{x,t}, \qquad y_{t+1} = y_t + a_{y,t}. \tag{6}$$

The policy is represented in the parameters $\theta$, and the result of applying the policy then the environment update is our function $f_\theta$.

Here, we consider verifying an agents trained with Deep-Q learning in an environment with $T_{\text{recharge}} = 3$ and $T_{\text{dirt}} = 4$, i.e, every cell accumulates dust four time-steps after it was cleaned, and the robot needs to recharge itself every 4 time-steps. Additionally, the initial cell where the robot starts can have between $0 - 0.1\%$ uncertainty in the amount of dirt, the charge that can be acquired from the recharge station, and the initial battery (as another element of uncertainty in the initial position).

If the agent leaves the domain, it is clipped back into the problem domain. The agent's each have 4 discrete actions and are trained with a combination of vanilla Deep-Q learning, Deep-Q learning + reward-shaping, and Deep-Q learning + verifiable training. The actions corresponds to velocities in the 4 cardinal directions, i.e., $\{(0, 5), (5, 0), (-5, 0), (0, -5)\}$. For the agent trained to be verifiable, in addition to loss from Deep-Q learning, the agent is also trained to be verifiable with respect to the temporal specification presented in Section 3.2.2. The verification losses are optimized using the Adam optimizer Kingma & Ba (2015) with learning rate $10^{-3}$. The weight of the verification loss anneals linearly between 0 and 1.5 during the first 70K steps. The model is trained to be verifiable starting at the center of the grids, and eventually covering the region around the centers, with $\epsilon = 0.015$ during the same 70k steps. We find that this regularizes the model to be verifiable in regions outside the region in which the model was trained to be verifiable.

## G    SHAKESPEARE

For the text generation task, we train a "seq2seq" model of Wu et al. (2016) with a GRU core. We use the standard test-train split for this task. The reconstruction and verification losses are optimized using the Adam optimizer (Kingma & Ba, 2015) with an initial learning rate of $10^{-1}$. The learning is decayed every 1K steps by a factor 10, till it reaches 0.001. The weight of the verification loss which is computed from the specification in Section 3.2.3 varies linearly between 20 and 1 during 100000 steps. Finally, the verification loss is clipped between below 10. The gradients during optimization are clipped at 0.1 to prevent exploding gradients from the differentiable approximations of `one-hot(softmax(.))`

### G.1    PREFIX TEMPLATE

The Language Model prefixes are generated using the following syntax: `<pronoun>, <person>, <action-verb>, <connector>, <person>, <pronoun>, <action-verb>` , where:

`<pronoun>` = {*'my', 'your', 'his', 'her', 'our', 'their'*}

`<person>` = {*'sister', 'brother', 'father', 'mother', 'son', 'daughter', 'king', 'queen', 'knight', 'noble','lord','duke', 'duchess', 'cousin', 'palace', 'widow','nurse', 'marshal', 'archbishop', 'mayor', 'maid'*}

`<action-verb>` = {*['changed','despised', 'loved', 'married', 'accused', 'anointed', 'danced', 'rejoiced', 'killed', 'came', 'left', 'prayed', 'stood', 'read', 'consorted', 'denied', 'condemned', 'ruled', 'proved', 'parted' 'resolved', 'committed', 'raised', 'urged', 'painted', 'provoked', 'lived', 'charged', 'yielded', 'accursed', 'assured'],* }

`<connector>` = {*'but', 'while', 'yet', 'and', 'because']*}

The space of combinations holds 25779600 possibilities to condition the language model generation upon. An example prefix is :'*Our lord yielded and their king left*'.

## H    Bound Propagation Through $f$

We describe how to perform bound propagation for a general recurrent neural network. The neural network takes as input $x$ and produces a sequence of outputs $y_\tau$ for $\tau = 0, ..., K$ so the overall output is $(y_0, y_1, \ldots, y_K)$. We assume that we are given bounds on the input $x$

$$l_0 \leq x \leq u_0.$$

Our goal is to obtain bounds on $y_\tau$ given bounds on $x$ for each $\tau$. Each output is produced conditioned on the preceding outputs: $y_\tau$ depends on $y_0, \ldots, y_{\tau-1}$.

We proceed recursively, assuming that we have already computed bounds on $y_0, \ldots, y_{\tau-1}$. We stack the set of inputs to the computation as $(x, y_0, \ldots, y_{\tau-1}) \in [l_{\tau;0}, u_{\tau;0}]$. We study the computation graph mapping these inputs to the output $y_\tau$. At each node in this computation graph, we perform a computation of the form

$$z_{\tau,i} = w_i^T h\left(z_{\tau;-i}\right) + \tilde{w}_i^T \tilde{h}\left(z_{\tau;-i}\right)^T z_{\tau;-i} + b_i$$

where $h, \tilde{h}$ are element-wise nonlinear operations (sigmoid, tanh, relu etc.) and $z_{\tau;-i}$ denotes the elements of computational graph that are ancestors of the node $i$. The second term represents multiplicative interactions (gating interactions) common in recurrent networks like LSTMs and GRUs. Suppose we have already computed lower and upper bounds $l_{\tau;-i}, u_{\tau;-i}$ on the preceding elements. Then, we have

$$z_{\tau,i} \geq \max\left(w_i, 0\right)^T h\left(l_{\tau;-i}\right) + \min\left(w_i, 0\right)^T h\left(u_{\tau;-i}\right)$$
$$+ 1^T \min\left( \begin{array}{cc} \tilde{w}_i \odot \tilde{h}\left(l_{\tau;i}\right) \odot \left(l_{\tau;i}\right), & \tilde{w}_i \odot \tilde{h}\left(u_{\tau;i}\right) \odot \left(l_{\tau;i}\right), \\ \tilde{w}_i \odot \tilde{h}\left(l_{\tau;i}\right) \odot \left(u_{\tau;i}\right), & \tilde{w}_i \odot \tilde{h}\left(u_{\tau;i}\right) \odot \left(u_{\tau;i}\right) \end{array} \right)$$
$$z_{\tau,i} \leq \max\left(w_i, 0\right)^T h\left(u_{\tau;-i}\right) + \min\left(w_i, 0\right)^T h\left(l_{\tau;-i}\right)$$
$$+ 1^T \max\left( \begin{array}{cc} \tilde{w}_i \odot \tilde{h}\left(l_{\tau;i}\right) \odot \left(l_{\tau;i}\right), & \tilde{w}_i \odot \tilde{h}\left(u_{\tau;i}\right) \odot \left(l_{\tau;i}\right), \\ \tilde{w}_i \odot \tilde{h}\left(l_{\tau;i}\right) \odot \left(u_{\tau;i}\right), & \tilde{w}_i \odot \tilde{h}\left(u_{\tau;i}\right) \odot \left(u_{\tau;i}\right) \end{array} \right)$$

Setting $l_{\tau,i}$ to the lower bound above and $u_{\tau,i}$ to the upper bound, we have computed bounds on $z_{\tau,i}$. Thus, we can recursively compute bounds until we obtain bounds $l_\tau \leq y_\tau \leq u_\tau$, which can then be used to compute bounds on $y_{\tau+1}$. Proceeding recursively, we obtain lower and upper bounds on $(y_0, y_1, \ldots, y_K)$.

## I    Shakespeare Text Generative Model Samples

| Method | Prefix | Generated Sequence |
|---|---|---|
| Verified Training | Very well; and | <eol> if we are like the king; and so the <eol> which of some other part |
| | Would you | <eol> could would be such other of the whole but is <eol> the a piece of |
| | If I must not, | <eol> as to be told general of the people, till he <eol> be the king to |
| | Soft! who comes | <eol> with the city but the whole prince and he <eol> are not them to the |
| Nominal Training | Very well; and | let <eol> 3 man <eol> will will not the new <eol> which the man of his |
| | Would you | <eol> shalt not the first to his face that you were not <eol> to the prince |
| | If I must not, | <eol> thing the matter but the man that he have <eol> not a piece of the |
| | Soft! who comes | <eol> these thing for the man that is the king <eol> <eol> is no matter of |

Table 7: Samples of generated text for nominal and verifiably training GRUs on the Shakespeare corpus under greedy decoding. <eol> refers to the end of line.

