# OpenReview forum: "Scalable Neural Learning for Verifiable Consistency with Temporal Specifications"
_ICLR.cc/2020/Conference — Reject_

### Official Review · AnonReviewer1 · 2019-10-23
**Official Blind Review #1**

**Rating:** 6

**Review:**

This paper concerns verification of neural networks through verified training and interval bound propagation. Namely, the authors rely on the fact (reported in the literature earlier, but also confirmed here) that verified training leads to neural networks that are easier to verify. The main contributions of this work are 1) extending interval bound propagation to recurrent computation and auto-regressive models as often encountered in NLP or RL settings which allows verified training of these models, 2) introducing the Signal Temporal Logic (STL) for specifying temporal constraints for a model, and extending its quantitative semantics for reasoning about sets of inputs, 3) providing empirical proof that the STL with bound propagation can be used to ensure that neural models conform to temporal specification without large performance losses.

The introduced method is well motivated and well-placed within the literature, with the related works section providing a good overview of the field; it also clearly mentions how the proposed method differs from the prior art. Section 3 describes the STL syntax and three specifications for different tasks; while the provided examples are nice, they are also long-winded and make the exposition difficult to follow---I think that their details should be moved to the appendix. Section 4 is very technical, and I do not have enough knowledge to verify it thoroughly, but the proposed approach seems to make sense. Finally, Section 5 presents experimental evaluation, which is performed on three different tasks: image caption generation, RL with a mobile robot, and language generation. These three experiments seem to be enough variety to prove the utility of the method. My only concerns are that 1) the loss of perplexity with verified training in the language modelling setup is disturbingly high as compared to the nominal method, and 2) the proposed method is compared to only one baseline in each experiment, and it is unclear whether the baseline are state-of-the-art without knowing the literature (which I do not know).

I recommend ACCEPTing this paper, albeit with low confidence. This is because the paper addresses an interesting and important problem, and the provided results are convincing. Having said that, I know nothing about the area of formal verification.

**Experience Assessment:**

I do not know much about this area.

**Review Assessment: Checking Correctness Of Derivations And Theory:**

I assessed the sensibility of the derivations and theory.

**Review Assessment: Checking Correctness Of Experiments:**

I assessed the sensibility of the experiments.

**Review Assessment: Thoroughness In Paper Reading:**

I read the paper at least twice and used my best judgement in assessing the paper.

---

> ### Author Response · Authors · 2019-11-13
> **Response to Reviewer 1**
>
> We thank the reviewer for the helpful comments. We have incorporated the suggestions regarding restructuring Section 3. We address the other points below.
>
>       1. We have included language samples from the different types of training (Appendix I).
>
> The perplexity is exponentiated negative log likelihood (NLL). Here, NLL is the task objective and on the test, the mean NLL is around 5.1, 5.2, and 5.4 for nominal, sampling and verified training respectively. In this regard, the degradation does not compromise the language model.
>
>       2. Baselines:
>
> We consider task-specific architectures as baseline models. Since this is the first work that uses STL based verified training of auto-regressive neural networks, there are no other verified training approaches to compare with. Nevertheless, we compare against the other prominent approaches in the literature for improving or verifying robustness of neural networks such as adversarial training and also consider the only currently available method for verifying auto-regressive networks with the MILP based approach from Wang et al., CVPR’19 .

---

> > ### Comment · AnonReviewer1 · 2019-11-14
> > **Maintin my score**
> >
> > Thanks for the response. I do not know much about the area and I feel that I am unable to properly evaluate this paper. I keep my recommendation of accepting this paper, although my confidence is low.

---

### Official Review · AnonReviewer3 · 2019-10-25
**Official Blind Review #3**

**Rating:** 8

**Review:**

This paper presents a way to train time-series regressors verifiably with respect to a set of rules defined by signal temporal logic (STL). The bulk of work is in deriving bound propagation rules for the STL language. The resulting lower bound of an auxiliary quantity (which is required to be non-negative) is then maximized for verifiability.

This technique is demonstrated on three tasks and compares favorably to the baseline from Wang et al. (2019).

I am not an expert in this area. However, to the best of my knowledge I don't see anything immediately wrong with this and it seems novel. Therefore I recommend acceptance.

The paper is also above recommended length.

**Experience Assessment:**

I do not know much about this area.

**Review Assessment: Checking Correctness Of Derivations And Theory:**

I did not assess the derivations or theory.

**Review Assessment: Checking Correctness Of Experiments:**

I assessed the sensibility of the experiments.

**Review Assessment: Thoroughness In Paper Reading:**

I read the paper at least twice and used my best judgement in assessing the paper.

---

> ### Author Response · Authors · 2019-11-13
> **Response to Reviewer 3**
>
> We thank the reviewer for the helpful comments. In order to improve accessibility of the paper, we did not want to assume specific background from the readers. This resulted in longer length of the paper. We have now moved some of the bound propagation details, and the task description details to the appendix along with a rewrite of Sections 2, 3, 4 to shorten the paper to recommended 8 pages.

---

### Official Review · AnonReviewer2 · 2019-10-26
**Official Blind Review #2**

**Rating:** 3

**Review:**

This paper extends bound propagation based robust training method to complicated settings where temporal specifications are given. Previous works mainly focus on using bound propagation for robust classification only. The authors first extend bound propagation to more complex networks with gates and softmax, and designed a loss function that replies on a lower bound for quantitative semantics of specifications over an input set S. The proposed framework is demonstrated on three tasks: Multi-MNIST captioning, vacuum cleaning robot agent and language generation. The authors formulate specifications using temporal logic, train models with bound propagation to enforce these specifications, and verify them after training.

My questions regarding this paper are mostly on the three demonstrated tasks:

1. For the Multi-MNIST dataset, I have the following questions:

1(a). In Table 2, it is surprising that even with perturbation of 0.5, the verified and adversarial accuracy is very high. At perturbation epsilon=0.5, it should be possible to perturb the entire image to gray (value 0.5), so I believe the accuracy should be very low here. It is hard to believe under this setting the verified accuracy is still 99%.

1(b). For Table 3, nominal accuracy should also be reported.

1(c). Additionally, how do you define the nominal accuracy here? An example is nominally correct when all digits are predicted correctly in the sequence, or just when the sequence length is predicted correctly?

1(d). For the termination accuracy, do we only care about the sequence length being predicted correctly, or does it also cover the case that all digits in the sequence are predicted correctly? If it is only concerning about the sequence length, this property is a little bit weak.

2. For the RL Robot agent experiment, I have the following questions:

2(a). Because T=10, are you saying that we can only guarantee that the battery is always recharged for any rollouts less than 10 steps? After 10 steps beyond the initial position, can we get any guarantees? A 10-step only guarantee seems too restrictive.

2(b). Are all the properties only verified assuming that the agent starts from the center? I think this assumption is probably also too strong in practice.

2(c). Since all the %verified cells reported in Table 4 are all 100%, it is probably better to make the problem more challenging, by increasing T or considering different initial positions. It is important to show when the performance of the proposed method starts to degrade, to understand the power of the proposed method.

3. For the language generation experiment, the perplexity of the verified training model looks significantly worse than nominal or sampled models. With a perplexity as high as this, I believe the model actually produces garbage. Can you provide some examples of generated texts? I feel language generation is probably not a suitable task for the proposed training method.

Other minor issues:

1. Table 1 should have some horizontal lines - it is hard to align the works with categories on the right.

2. Several papers appear multiple times in references, including "Differentiable abstract interpretation for provably robust neural networks", "Towards fast computation of certified robustness for relu networks" (and probably others). Also on page 2, Shiqi et al., should be Wang et al. (Shiqi is the first name).

3. I feel the writing is a bit rushed and the authors should make a few more passes on the paper.


This paper makes valid technical contributions, especially the conversion from STL specifications to lower bounds of the quantitative semantics is interesting. Although bound propagation based robust training method is simple to extend to softmax/GRU with interval analysis, applying robust training techniques to the three interesting applications are good contributions. Since the main contribution of this paper is the empirical results on the three tasks, my concerns regarding the experiments need to be addressed before I can vote for accepting this paper. Also, because this paper uses 10 pages, I am expecting the paper to meet a higher standard. Thus, I am voting for a weak reject at this time.


****** After author response

The author response addressed some of my concerns. However I do believe this paper is relatively weak in contribution, especially the experiments can be done more thoroughly. I also appreciate that the authors reduced paper length to 8 pages. I am okay with accepting this paper as it does have some interesting bits, but it is clearly on the borderline and can be further improved.


**Experience Assessment:**

I have published in this field for several years.

**Review Assessment: Checking Correctness Of Derivations And Theory:**

I assessed the sensibility of the derivations and theory.

**Review Assessment: Checking Correctness Of Experiments:**

I carefully checked the experiments.

**Review Assessment: Thoroughness In Paper Reading:**

I read the paper at least twice and used my best judgement in assessing the paper.

---

> ### Author Response · Authors · 2019-11-13
> **Response to Reviewer 2**
>
> We thank the reviewer for the detailed and helpful comments. We have improved the presentation of the experimental results as suggested by the reviewer. We have also reduced the length of the paper. We respond to the comments in details below.
>
> 1(a). Note that the specification we consider from Wang et al., CVPR’19  is only violated when the network produces a longer sequence than the true number of digits in the input image. At large perturbations (eps ~ 0.5), when the digits become indistinguishable, the network learns to predict that the image has no more than one digit. This satisfies the specification (we make this observation at the end of page 8 in the original submission, have clarified further during revision -- page 7, paragraph above run-time considerations), even if the nominal accuracy is very low. This is in contrast with standard adversarial robustness, where at large perturbations the epsilon balls of images from different classes start to overlap and it is not possible to be robust, resulting in low verified accuracies.
>
> 1(b). We have added nominal accuracy in Table 3 (Caption, Table 3).
>
> 1(c). We use token-level accuracy as a measure of nominal accuracy, that is, the percentage of correctly predicted tokens -- including the end of sequence token.
>
> 1(d). By definition, termination accuracy only covers sequence length. This specification is motivated by Wang et al., CVPR’19 where it was shown that sequence generating models are often vulnerable to violating this property. Nevertheless, richer temporal properties can be specified in STL and analyzed through our method.
>
> 2(a). We guarantee that at every time-step within the first T steps, the battery is recharged within the next T_recharge steps. For experimentation purposes, we use T=10, but it can be configured. In this work, we focus on the core problem of training neural networks in a verified manner for a finite number of time steps.
>
> It is worth noting in this context that verifiably trained policies satisfying such bounded-time properties can be chained to plan for infinite time horizons in a receding horizon manner with on-line monitoring. For instance, in a scenario where the agent has bounded uncertainty on it’s pose at each time instant, we can monitor online to guarantee that there is no failure in the next T steps at every time instant of an infinite horizon execution (a similar approach is used in https://arxiv.org/abs/1703.09563 ).
>
>
>
> 2(b). No, we don’t assume that the agent starts at the center. The agent starts in an epsilon-neighborhood of the center of any cell on the map (Figure 3). We’ve reworded Appendix F1 to make this more clear.
>
> 2(c). Different initial positions would not degrade the performance with regard to verification guarantees we are able to provide, assuming the new initial positions are considered during verified training. The two key things that would cause a drop in verification performance are i) the size of the initial region, ii) the time horizon.
>
> As we increase the size of initial regions we are verifying, due to the increased variations, the performance can degrade, but the verification approach still achieves better results. At \epsilon=2.0, the fraction of cells for which we can verify the property drops to ~64% while that for the baseline approaches drops to under 20%. Further, depending on the coefficient for the verification loss, even for longer horizon problems, the verification performance might stay similar but with a possible reduction in the reward. We will consider including experiments with longer horizons for the final version.
>
> 3. The text generation model is able to produce reasonable text. We have included some samples (Appendix I). The perplexity is exponentiated negative log likelihood (NLL). Here, NLL is the task objective and on the test, the mean NLL is around 5.1, 5.2, and 5.4 for nominal, sampling and verified training respectively. In this regard, the degradation does not compromise the text generating model.
>
> Minor issues and presentation length:
> We have fixed the minor issues, thanks for pointing them out. In order to improve accessibility of the paper, we did not want to assume specific background from the readers. This resulted in longer length of the paper. We have now moved some of the bound propagation details to the appendix  and restructured the  task description details to shorten the paper to recommended 8 pages.

---

> ### Author Response · Authors · 2019-11-14
> **Thank you for the detailed feedback!**
>
> Dear Reviewer,
>
> Thank you for the detailed review, and feedback. Please let us know if our response addresses your concerns about the empirical results. We would be happy to provide additional details if required.

---

### Official Review · AnonReviewer5 · 2019-10-31
**Official Blind Review #5**

**Rating:** 1

**Review:**

This paper focuses on verifying sequential properties of deep beural networks. Linear Temporal Logic (LTL) is a
natural way to express temporal properties, and has been extensively studied in the formal methods community.
Signal temporal logic (STL) is a natural extension,  of LTL. STL specifications provide a rich set of formulations to encode intent for real valued signal over time. Formally proving STL formulae is intractable. But, it is possible to falsify such properties. This has been the main goal for various tools like Breach, and S-Taliro.


Pros :
A  very  interesting avenue  explored in this paper, is using the syntax of
STL to formulate properties about multiple-MNIST, Safe RL and NLP applications.
Even though the conversion from STL specifications to scalar valued function is a very well known technique.

Cons :
In my opinion, the paper lacks sufficient contributions in itself to be accepted at this conference. The idea of training
for robustness using intervals, has been well known for a while. The authors extend that to get conservative estimates of the level of satisfaction of the STL formula, and use that in the training process. Though training for robustness is an
interesting idea in itself, but the general opinion about using interval propagation to train networks is negative.

Overall : Though the direction of this work is interesting but lacks sufficient technical novelty.

**Experience Assessment:**

I have published one or two papers in this area.

**Review Assessment: Checking Correctness Of Derivations And Theory:**

I assessed the sensibility of the derivations and theory.

**Review Assessment: Checking Correctness Of Experiments:**

I assessed the sensibility of the experiments.

**Review Assessment: Thoroughness In Paper Reading:**

I read the paper thoroughly.

---

> ### Author Response · Authors · 2019-11-13
> **Response to Reviewer 5**
>
> We thank the reviewer for the helpful comments.
>
> >>Response to “lack of sufficient contributions”
>
> STL has been shown to be an expressive formalism to specify temporal properties of real-valued signals. As the reviewer points out, it has been used in detecting violations of specifications (falsification) in the hybrid systems domain by approaches like S-Tarilo and Breach. The novel  contributions of our paper are as follows:
> We make a novel connection between STL and verified training of neural networks that generate sequential outputs which facilitates training networks that can be easily verified to be consistent with an STL specification. Unlike falsification which typically employs stochastic search for falsifying inputs, we design a novel loss function derived by computing a differentiable bound on the worst-case violation of the STL specification (under quantitative semantics) and incorporate this into an SGD-based training scheme for the network.
> We demonstrate the practical utility of this approach on neural networks (including auto-regressive RNN/GRUs) for captioning, language generation and reinforcement learning -- tasks and architectures that have not been considered for verified training before.
>
> We have added a discussion to relate our work with S-Tarilo and Breach in Section 2 (first paragraph, page 3).
>
> >>Response to “general negative opinion about the use of interval propagation to train neural networks”
>
> We respectfully disagree with the reviewer’s claim. On the contrary, [1, 2] have shown that interval propagation can be used very effectively to verifiably train large neural networks that were beyond the scope of other methods. Specific to our paper, we clearly demonstrate that interval propagation can be leveraged effectively to train complex networks such as RNNs and for complex temporal specifications. Very recent papers such as [3, 4] are some other examples of successful use of interval propagation for verified training of neural networks. Further, we note that the overall framework of training neural networks to be verifiable against STL that we present in this work can also be instantiated using any other method of over-approximating reachable states of neural networks (for example using LP relaxations or other abstractions).
>
> [1] On the Effectiveness of Interval Bound Propagation for Training Verifiably, Gowal, S. et al., ICCV’19
> [2] A Provable Defense for Deep Residual Networks. Mirman, M. et al., https://arxiv.org/abs/1903.12519
> [3] Certified Robustness to Adversarial Word Substitutions, Jia R. et al., EMNLP’19 https://arxiv.org/abs/1909.00986
> [4]  Achieving Verified Robustness to Symbol Substitutions via Interval Bound Propagation, Huang, P. et al., EMNLP’19 https://arxiv.org/abs/1909.01492

---

> > ### Comment · AnonReviewer5 · 2019-11-14
> > **Reply to Author's Response**
> >
> > The property of robustness to bounded perturbation to pixel values does not say much about the correctness
> > of the image classifiers. Having said that, it does prove robustness to some local perturbations. So training
> > against robustness seems like a high price to pay in training. In fact the perturbation epsilons
> > quickly start losing meaning, without an understanding of the image manifold.
> >
> > The contributions pertaining to robust training of neural networks presented in this paper, in my opinion,
> > is not significant enough  to be considered for publication in this conference. As I have mentioned before
> > the connection between STL and training is interesting, but does not meet the technical expectation of this
> > conference. It's a trivial interval adaption of the basic semantics of STL in SGD framework,  and I am not convinced that there is anything significant in that.
> >
> > I would stay with my suggestion to reject the paper.

---

> > > ### Author Response · Authors · 2019-11-14
> > > **Response**
> > >
> > > Dear Reviewer,
> > >
> > > We thank you for the follow up comments.
> > >
> > > >>The property of robustness to bounded perturbation to pixel values does not say much about the correctness of the image classifiers. Having said that, it does prove robustness to some local perturbations.
> > >
> > > Our work follows a long line of research motivated by the influential papers (e.g., [1], [2]) which demonstrated lack of robustness of neural networks to small, local perturbations. In particular, we endeavor to bring the tools of verification to address this problem. We specifically go beyond the robustness of classification in this paper, and express and verify temporal properties of sequence-producing neural networks. In addition to multi-MNIST captioning, we also present results on text generation and RL policy networks. For multi-MNIST, we consider bounded perturbation to pixel values. The termination property is motivated by the vulnerability of RNNs to generating longer sequences under small bounded perturbations [3]. Further, our work is not limited to pixel perturbations -- for text generation, we consider word substitutions (e.g., synonym substitutions) and for RL agents, we consider uncertainty in initial pose. Word substitutions are interesting and have been studied in the classification setting as well [4].
> > >
> > > >> So training against robustness seems like a high price to pay in training.
> > >
> > > Training for robustness is performed efficiently with only additional cost of 2x slow down (Section 4.2). Further we also show that it does not result in significant degradation of the task objective (Section 5).
> > >
> > > >>As I have mentioned before the connection between STL and training is interesting, but does not meet the technical expectation of this conference. It's a trivial interval adaption of the basic semantics of STL in SGD framework,  and I am not convinced that there is anything significant in that.
> > >
> > > We believe that this connection between STL and verified training is important. While simple, we show that this is indeed effective in verified training for properties that go beyond misclassification. Further, we note that our contributions extend beyond this connection -- we extend worst-case analysis to novel architectures, and we also empirically demonstrate that these extensions are effective in practice (Section 5).
> > >
> > > [1] Intriguing properties of neural networks, Szegedy et al., 2014
> > > [2] Towards Evaluating the Robustness of Neural Networks, Carlini et al., https://arxiv.org/abs/1608.04644
> > > [3] Knowing When to Stop: Evaluation and Verification of Conformity to Output-size Specs, Wang et al.,  CVPR’19
> > > [4] Certified Robustness to Adversarial Word Substitutions, Jia R. et al., EMNLP’19 https://arxiv.org/abs/1909.00986

---

### Author Response · Authors · 2019-11-13
**Revised manuscript**

We thank the reviewers for the feedback. We have responded to their comments below.

We have revised the paper to address the comments from the reviewers. The main changes to the paper are:
1) An improved experiments section to address Reviewer 2’s comments
2) Shortening of the paper to address the comment about the paper being longer than the recommended length from Reviewers 1 and 3.
3) We have also restructured Section 3 as suggested by Reviewer 2, and moved some of the details to the Appendix.

We would be happy to answer any further queries.

---

### Decision · Program_Chairs · 2019-12-19

**Decision:**

Reject

**Comment:**

This submission proposes a deep network training method to verify desired temporal properties of the resultant model.

Strengths:
-The proposed approach is valid and has some interesting components.

Weaknesses:
-The novelty is limited.
-The experimental validation could be improved.

Opinion on this paper was mixed but the more confident reviewers believed that novelty is insufficient for acceptance.